# Platelet Activation in Ovarian Cancer Ascites: Assessment of GPIIb/IIIa and PF4 in Small Extracellular Vesicles by Nano-Flow Cytometry Analysis

**DOI:** 10.3390/cancers14174100

**Published:** 2022-08-24

**Authors:** Barbara Bortot, Alessandro Mangogna, Ben Peacock, Rebecca Lees, Francesco Valle, Marco Brucale, Sara Tassinari, Federico Romano, Giuseppe Ricci, Stefania Biffi

**Affiliations:** 1Obstetrics and Gynecology, Institute for Maternal and Child Health, IRCCS Burlo Garofolo, 34137 Trieste, Italy; 2NanoFCM Co., Ltd., MediCity, Nottingham NG90 6BH, UK; 3Consorzio per lo Sviluppo dei Sistemi a Grande Interfase (CSGI), 50019 Sesto Fiorentino, Italy; 4Consiglio Nazionale delle Ricerche, Istituto per lo Studio dei Materiali Nanostrutturati (CNRISMN), 40129 Bologna, Italy; 5Department of Medical, Surgical and Health Science, University of Trieste, 34127 Trieste, Italy

**Keywords:** ovarian cancer, ascites, small extracellular vesicles, platelet, nano-flow cytometry, GPIIb/IIIa, PF4

## Abstract

**Simple Summary:**

Platelets play a critical role in coagulation and fibrinolysis processes, but recent literature also indicates their central involvement in immune response, cancer progression and metastasis. During platelet activation, small extracellular vesicles (EVs) are released. The ascites are a fluid developing in the peritoneum of ovarian cancer patients in an advanced stage. This study analysed the expression of platelet markers GPIIb/IIIa and PF4 in small-EVs populations isolated from the ascitic fluid of patients with advanced ovarian cancer. The percentage of platelet-derived small-EVs was positively correlated with platelet distribution width to platelet count in sera (PDW/PLT), a surrogate indicator of platelet activation. Overall, we presented a method that can be helpful in future studies to determine the correlation between the extent of platelet activation in ascites and disease status.

**Abstract:**

In ovarian cancer, ascites represent the microenvironment in which the platelets extravasate to play their role in the disease progression. We aimed to develop an assay to measure ascites’ platelet activation. We enriched small extracellular vesicles (EVs) (40–200 nm) from ascites of high-grade epithelial ovarian cancer patients (*n* = 12) using precipitation with polyethylene glycol, and we conducted single-particle phenotyping analysis by nano-flow cytometry after labelling and ultra-centrifugation. Atomic force microscopy single-particle nanomechanical analysis showed heterogeneous distributions in the size of the precipitated particles and their mechanical stiffness. Samples were fluorescently labelled with antibodies specific to the platelet markers GPIIb/IIIa and PF4, showing 2.6 to 18.16% of all particles stained positive for the biomarkers and, simultaneously, the EV membrane labelling. Single-particle phenotyping analysis allowed us to quantify the total number of non-EV particles, the number of small-EVs and the number of platelet-derived small-EVs, providing a platelet activation assessment independent of the ascites volume. The percentage of platelet-derived small-EVs was positively correlated with platelet distribution width to platelet count in sera (PDW/PLT). Overall, we presented a high-throughput method that can be helpful in future studies to determine the correlation between the extent of platelet activation in ascites and disease status.

## 1. Introduction

Ovarian cancer consists of a heterogeneous group of neoplasms, of which about 90% are epithelial (mucinous, serous, endometrioid, and clear cell subtypes) [1]; most women are diagnosed at an advanced stage of disease with a high-grade serous ovarian cancer subtype (HGSOC). HGSOC is significantly more aggressive than low-grade serous ovarian cancer and is associated with a poorer prognosis [2]. The lack of screening tools and early diagnosis methods substantially contributes to the unfavourable prognosis, suggesting a need to identify effective biomarkers for HGSOC.

The ascites are a fluid developing in the peritoneum of HGSOC patients in an advanced stage, and its cellular and extracellular vesicle (EV) composition can reveal the inflammatory environment state of the malignant disease [3]. The evidence that tumour cells release more EVs than healthy cells points to their role in tumour progression and the potential of EVs-based liquid biopsy as diagnostic biomarkers in ovarian cancer [4].

Platelets are anucleate cell fragments formed and released into the bloodstream from bone marrow megakaryocytes (MK) [5]. Platelets serve as major contributors to haemostasis, which can be subdivided into primary haemostasis, secondary haemostasis and fibrinolysis [6]. Platelets are activated upon contact with extracellular matrix proteins, including collagen, von Willebrand factor, and fibronectin, in response to vascular injury. During primary haemostasis, platelet adhesion at a site of vascular damage entails a collaborative effort of various platelet receptors, leading to platelet activation and aggregation. In secondary haemostasis, the formation, deposition and cross-linking of insoluble fibrin, generated by the coagulation cascade, stabilize the primary platelet plug [6]. The highly regulated enzymatic process of fibrinolysis prevents unnecessary accumulation of intravascular fibrin. 

Besides their haemostatic function, platelets play two major non-haemostatic roles related to the immune response: they have an anti-infective nature and influence innate and adaptive immune systems [7]. Platelets can communicate with numerous immune cells such as B-cells, T-cells, macrophages, neutrophils, EC, natural killer cells and DC, leading to a spectrum of immune-related events [7]. 

Platelets are recognized as pivotal players in numerous other processes, and recent literature indicates their central involvement in cancer progression and metastasis [8,9]. The role of platelets in the formation of the early metastatic niche can be divided into three major phases: (i) the formation of a tumorigenic microenvironment by deregulation of extracellular matrix (ECM) dynamics and the recruitment of granulocytes; (ii) the promotion of tumour-associated angiogenesis (an essential aspect of cancer progression as tumour cells face increasing demand for nutrient, oxygen, and metabolic waste exchange; furthermore, tumour vasculature is the main route through which cancer cells metastasize and immune cells infiltrate), and (iii) damping of the immune surveillance by the induction of an immune-suppressive environment in the metastatic niche [10].

Biochemical and cellular evidence has pointed out that complex bidirectional interactions between platelet and cancer cells in the tumour microenvironment, bloodstream and peritoneal fluid are essential for its progression and metastasis [11,12]. Platelet-cancer cell interaction is critical in cancer cells’ ability to evade detachment-induced apoptosis (a process commonly referred to as anoikis), a major metastasis hallmark [12]. Recent results demonstrated that platelets educate tumour cells by highly efficient transference of lipids, proteins and RNA through different mechanisms, strengthening the notion that tumour cells might acquire aggressive phenotypes due to platelet interaction [13].

Specifically, in ovarian cancer, preclinical models convincingly support that platelets extravasate from the circulation into the tumour microenvironment, increase the proliferation of neoplastic cells and promote epithelial to mesenchymal transition [14,15,16,17]. A study by Saha et al. recently described an ovarian tumour microenvironment chip that enabled imaging of the complex interaction between the ovarian cancer cells and blood platelets and the resulting metastatic process [18]. In another work, a multi-cellular model based on HGSOC cell lines enabled the investigation of platelets’ role in malignant cell invasion and ECM production [19].

Only a little research has been conducted on clinical samples to investigate the contribution of platelets to the biological characteristics of micrometastases. However, anti-platelet therapy was tested in patients based on the crucial contribution of platelet to the formation and expansion of the early metastatic niche [10]. Furthermore, platelet counts and activation markers significantly impact cancer prognosis and predict treatment response [9,20]. Platelet-to-lymphocyte ratio (PLR) is often associated with inflammation, and it has been suggested as a prognostic factor in various cancers [21]. In this respect, further prospective-control studies in different clinical settings will help clarify this hypothesis. Interestingly, cancer treatment regimens, including surgery, may also increase the risk of hypercoagulability in these patients through impact on platelet activity. Paraneoplastic thrombocytosis in ovarian cancer is associated with advanced disease and shortened survival. Among clinical trials on the risk of thrombotic events and venous thromboembolism in cancer patients receiving chemotherapy, one study reported an increased chance of thrombosis by 6- to 7-fold [22].

During platelet activation, two distinct EV populations are released: a part of EVs derived from the plasma membrane (microvesicles) and a separate one of exosomes that are small EVs [23]. Several potential markers were identified on platelet-derived small-EVs, with the expression of CD31, CD41, CD42a, P-selectin, PF4, and GPIIb/IIIa detected besides markers of EVs [24,25]. We herein hypothesized that platelets are activated upon extravasation in ascites and release small-EVs that can be quantified relative to the small-EVs total population by single-particle phenotyping analysis, thus enabling individual patient scores. To explore this hypothesis, we performed nano-flow cytometry analyses by measuring two platelet-associated markers, GPIIb/IIIa and PF4, in ascites-derived small-EVs. Pre-and post-chemotherapy ascites samples were obtained from the same ovarian cancer patients. Overall, our results provide for each sample the total number of non-EV particles, the number of small EVs and the number of platelet-derived small EVs, thus assessing the extent of platelet activation in ascites. We present a high-throughput method that can be helpful in future studies to determine the correlation between the extent of platelet activation in ascites and disease status. 

## 2. Materials and Methods

### 2.1. Patients’ Cohort and Samples Collection

We conducted a prospective analysis of patients with advanced HGSOC (histological grade G3 confirmed with histological examination, definitive analysis, or frozen section) with bulky stage IIIB-IIIC to IV disease (FIGO stage classification) treated at our Institution.

Based on the intraoperative clinical evaluation performed during diagnostic laparoscopy to determine the likelihood of resectability [26], patients underwent primary debulking surgery (PDS) or neoadjuvant chemotherapy followed by interval debulking surgery (IDS). In addition, they were asked to sign an informed consent form, following approval by the regional ethics committee of Friuli Venezia Giulia, Italy (CEUR); protocol number 4829.

### 2.2. Haemoglobin Detection 

Haemoglobin concentration was assessed using a Haemoglobin Assay Kit (Abcam, ab234046, Boston, MA, USA). Briefly, 20 µL of blood was incubated with 180 µL of Haemoglobin Detector solution at room temperature for 15 min on a 96-well plate. The absorbance was measured at 575 nm employing a GloMax^®^ Discover Microplate Reader instrument (Promega, Madison, WI, USA). The intensity of the colour was directly proportional to the sample’s haemoglobin concentration, which was determined by performing a Haemoglobin standard curve ranging from 0–250 mg/dL.

### 2.3. Small-EVs Isolation

Small-EVs from ascites were isolated as previously described [27], using Total Exosome Isolation Reagent (Invitrogen (Waltham, WA, USA), CN 4484453), following the protocol specified by the manufacturers. 1 mL of sample (ascites fluid) was centrifuged at 2000× *g* at room temperature to remove cells and debris. The supernatant containing the clarified fluid was transferred to a new tube without mixing up the pellet. 0.5 mL of Total Exosome Isolation Reagent was added. The sample was mixed by vortexing and incubated at room temperature for 30 min. After incubation, the sample was centrifuged at 10,000× *g* for 10 min, and the supernatant was discarded. The pellet was resuspended by adding 500 μL 1 × PBS.

### 2.4. Atomic Force Microscopy (AFM) Imaging and Morphometric Analysis 

AFM imaging was performed on glass coverslips coated with Poly-L-lysine (PLL) prepared as follows. Microscopy glass slides (15 mm diameter round coverslips, Menzel Gläser) were incubated for 120′ in a 3:1 (*v*:*v*) 96% H_2_SO_4_/30% H_2_O_2_ ‘piranha’ solution, rinsed with ultrapure water, sonicated (Elmasonic Elma S30H) for 30′ in acetone, 30′ in isopropanol and 30′ in ultrapure water, then cleaned with an air plasma (Pelco EasiGlow) for 5′ and immediately immersed in ultrapure water. Clean slides were then incubated for 30′ in a freshly prepared 0.01 (mg/mL) PLL solution, thoroughly rinsed with ultrapure water and dried with a gentle nitrogen flow.

Samples were left to adsorb on functionalized substrates for 30′ at 4 °C, then inserted into the AFM fluid cell. Before imaging, ultrapure water was briefly fluxed into the fluid cell to remove unabsorbed specimens. The concentration of each sample was adjusted by trial and error in successive depositions to maximize the ratio of isolated objects, thus facilitating their morphometric analysis.

All AFM images were taken in PeakForce mode on a Bruker Multimode8 equipped with a Nanoscope V controller, a sealed fluid cell and a type JV piezoelectric scanner using Bruker ScanAsystFluid+ probes (triangular cantilever, nominal tip curvature radius 2–12 nm, nominal elastic constant 0.7 N/m) calibrated with the thermal noise method [28]. Image analysis was performed with a combination of Gwyddion 2.58 [29] and custom Python scripts to recover the surface contact angle and equivalent solution diameter of individual objects [30].

### 2.5. Nano-Flow Cytometry 

A NanoAnalyzer U30 instrument (NanoFCM Inc., Nottingham, UK) equipped with dual 488/640 nm lasers and single-photon counting avalanche photodiode detections (SPCM APDs) was utilized for simultaneous detection of side scatter (SSC) and fluorescence of individual particles. Implemented bandpass filters let us catch the light in specific channels (SSC—488/10; FL1—525/40; FL2—670/30). Sheath-fluid of HPLC grade water was gravity fed, focusing the sample core stream diameter to ~1.4 µm. Measurements were taken over 1-min durations at a sampling pressure of 1.0 kPa, maintained by an air-based pressure module. All samples were diluted to achieve a particle count within the optimal range of 2000–12,000/min. During sample acquisition, the sample stream was fully illuminated within the central region of the focused laser beam, resulting in approximately 100% detection efficiency and accurate particle concentration measurement via single-particle enumeration.

Particle concentrations were determined against a standard of 250 nm silica nanoparticles of known concentration. EV isolates were sized according to standard operating procedures using a proprietary 4-modal silica nanosphere cocktail (NanoFCM Inc., S16M-Exo) with diameters of 68, 91, 113 and 155 nm. A standard curve was generated based on the side scattering intensity of the four different silica particle sub-populations using the NanoFCM software (NanoFCM Profession V1.8). Silica provided a stable and monodisperse standard with a refractive index of approximately 1.43 to 1.46, which was close to the range of refractive indices reported in the literature for EVs (*n* = 1.37 to 1.42). The laser was set to 10 mW and 10% SSC decay.

For labelling of EVs, the following antibodies were used: anti-PF4 conjugated to FITC (Polyclonal Rabbit anti-Human CXCL4/PF4 Antibody (FITC) LS-C670941, LSBio, Seattle, WA, USA), and anti-GPIIb/IIIa FITC (FITC mouse anti-human PAC-1 code 340507, BD Biosciences, Franklin Lakes, NJ, USA). The dye CellMask™ Deep Red Plasma Membrane Stain (Thermo Fisher, Waltham, MA, USA) was also utilized. 10 µL incubations of EV sample and labelling reagents were prepared with particle concentrations, determined by unlabelled nFCM, of 2 × 10^10^ particles/mL (Dilutions: PF4—1:10, GPIIb/IIIa—1:10, CellMask—1:500) before incubation for 1 h at room temperature. After incubation, the mixture was diluted in PBS to 1 mL to be pelleted by ultracentrifugation with a benchtop optima TLX (Beckman Coulter) for 1 h at 100,000× *g*. The supernatant was removed, and pellets were resuspended to 1 × 10^8^–1 × 10^9^ particles/mL for immediate nFCM analysis.

Data processing was handled within the nFCM Professional Suite v1.8 software, with dot plots, histograms, and statistical data provided in a single PDF. Gating within the software allowed the proportional analysis of subpopulations separated by fluorescent intensities with size distribution and concentration available for each sub-population [31,32].

### 2.6. Protein Expression Analysis by Western Blotting

The Western blotting analysis was performed using TSG101 (ABCAM, ab125011), CD63 (ABCAM, ab213090) and anti-Platelet IIb/IIIa complex (85/661) (sc-73544, Santa Cruz Biotechnology, Dallas, TX, USA) antibodies, starting from 30 μg of proteins extract-ed from ascites extracellular vesicles. We developed the membranes with the Claritymax Western ECL Substrate (Bio-Rad, Hercules, CA, USA) solution and captured the images with the ChemiDoc MP instrument (Bio-Rad, Hercules, CA, USA).

### 2.7. Statistical Analysis

Pearson’s correlation analysis was conducted with GraphPad Prism software version 9.0. Correlation coefficients (r) larger than zero displayed a positive relationship between the two selected data sets. *p*-value < 0.05 was considered as significant.

## 3. Results

### 3.1. Patients’ Selection and Clinical Data

The characteristics of the patients, classified as HGSOC following pathology review, are summarized in Table 1. Sixteen ascites samples were collected during laparoscopic surgery (asterisks indicate them in Table 1); eleven samples were collected during the exploratory laparoscopy performed for diagnostic purposes, and five samples after the completion of neoadjuvant chemotherapy at the time of the interval debulking (three months after the first sample).

PLR, derived from the absolute platelets and lymphocyte counts of a complete blood count, were previously described as available markers of the systemic inflammatory response [33]. 10 out of 12 patients presented values of PLR out of the range considered normal (60.0–239.0), suggesting a systemic inflammatory state. The ratio of platelet distribution width to platelet count (PDW/PLT) is a common indicator of platelet size and a surrogate indicator of platelet activation [34]. In a previous study, clinical data of 80 patients with serous ovarian cancer showed that the reduction of PDW/PLT was significantly related to an increase in the overall survival and the median progression-free survival [34]. According to this study, there were significant differences in FIGO stage, lymph node metastasis, and ascites between low PDW/PLT (<0.0845) and high PDW/PLT (>0.0845) groups. In our cohort, PDW/PLT values ranged from 0.013 and 0.046.

### 3.2. Haemoglobin Levels in Ascites Are below 0.5% of the Corresponding Sera Level

Ascites liquids presenting evidence of blood contamination were discharged and not included in the study. Haemoglobin levels were measured in ascites samples to determine the presence of blood contamination. Table 1 Supplementary shows the haemoglobin concentration values in ascitic fluids and the corresponding sera. For all ascites samples, the haemoglobin level was less than 0.5% of the mean value found in the sera, suggesting that no blood contamination in the ascites interfered significantly with the analysis.

### 3.3. Nanomechanical Properties of Small-EVs in Ascites Measured by Morphometric Analysis of AFM Images

To verify the presence of intact vesicles in small-EVs samples enriched from ascites isolates as detailed above (see “Small-EVs isolation” in the methods section), we recorded at least 5 AFM micrographs of each of 10 representative samples. Qualitative inspection of the AFM images confirmed the presence of abundant globular objects corresponding to putative EVs in all isolates (Figure 1A). The relative amounts of total particles found in each sample were estimated by calculating their surface density after substrate adsorption, revealing a substantial concentration heterogeneity across the panel. In general, however, specimens from the same patient showed an increased particle count (Figure 1C) in samplings at t = 1. EVs CD63 and TSG101 markers were assayed using Western blotting analysis (Appendix A). The size and mechanical stiffness of each individual object found in all micrographs were calculated via quantitative AFM morphometry, then plotted on diameter (D) vs. contact angle (CA) plots (Figure 1B) as described elsewhere [30]. The overall mechanical fingerprint of the particles found in each sample was obtained by pooling the measurements of between 76 and 927 individual objects.

Enriched particles were found to have diameters between 40 and 900 nm. In all samples, particles above ~150 nm showed the characteristic nanomechanical behaviour of intact vesicles [27,30,35], which results in horizontally elongated clusters of objects having similar CAs (and thus, mechanical stiffnesses). EVs with D values above 150 nm tended to cluster around different CA values in samples from different patients; interestingly, EVs enriched from the same patient tended to maintain the same average CA value even when collected at successive time points.

Particles with diameters < 150 nm were invariably more abundant than their larger counterparts; however, due to their small size, the former only accounted for between 18% and 58% of the total particle volume despite constituting 70–90% of the total number of particles. These smaller objects did not show the characteristic mechanical fingerprint of intact vesicles on their CA/D plots, leaving the possibility open that they might correspond to other classes of co-isolated biomaterials such as lipoproteins or protein aggregates.

### 3.4. Single-Particle Phenotyping Analysis of Small-EVs in Ascites Revealed Platelet-Derived GPIIb/IIIa and PF4 in All the Samples

As already reported in the literature, GPIIb/IIIa is an integrin complex that acts as a receptor for fibrinogen and vWf, leading to the adhesion of platelets to fibrinogen [36]. Western blotting analysis has verified this biomarker’s presence in the ascites-derived EVs (Appendix A). To quantify the platelet-derived EVs with a solid labelling procedure, we selected a second marker that is reported to be selectively present on the platelet-derived EVs [25]. We have analysed the presence of GPIIb/IIIa and PF4 in 10 representative samples, and we have compared the results with the single staining based on GPIIb/IIIa alone (Figure 2). Although the differences were not statistically significant, the highest positivity values were obtained when both these markers were used.

Subsequently, single-particle phenotyping analysis allowed the measurement of the platelet-associated markers GPIIa/IIIb and PF4 in all the samples, along with the EV membrane staining (Figure 3). The sizing profile of the samples analysed by nano-flow cytometry (NanoFCM) demonstrated that preparations upon labelling and ultracentrifuge were enriched in small-EVs (ranging from 40 to 200 nm), with a mean diameter range from 68 to 95 nm in size.

Samples exhibited heterogeneity in their concentration (Figure 4A). The analysis revealed that 20 to 63% of all detected particles were cell membrane positive (Figure 4B). Samples were fluorescently labelled with FITC-conjugated antibodies specific to the associated platelet markers GPIIb/IIIa and PF4. 2.6 to 18.16% of all particles stained positive for the platelet biomarkers and simultaneously the membrane labelling (Figure 4C). 4.6 to 21.2% of all cell membrane positive EVs stained positive for the platelet-associated biomarker (Figure 4D). Pearson’s correlation analysis proved that the percentage of particles stained positive for the platelet biomarkers relative to total particles was positively correlated with the percentage of particles stained positive for the platelet biomarkers relative to EVs (r = 0.7990, *p* = 0.0004) (Figure 4, lower graph).

### 3.5. The Percentage of Platelet Biomarkers Measured in Small-EVs from Ascites Was Positively Correlated with PDW/PLT in Sera

Pearson’s correlation analysis showed that the percentage of platelet biomarkers relative to total particles in ascites was positively correlated with PDW/PLT in sera (r = 0.6307, *p* = 0.0117) (Figure 5A). Also, the percentage of platelet biomarkers relative to EVs was positively correlated with PDW/PLT in sera (r = 0.5832 *p* = 0.0225). These significant positive correlations suggest that patients with a high percentage of platelet biomarkers in ascites will usually also display a higher PDW/PLT in sera and vice versa. The graphs in Figure 5A,B imply that patients who scored high on the platelet activation index in ascites tend to have high intravascular platelet activation scores. These complementary data sets from ascites might improve the patients’ classification to yield more robust representations. The two measures might complement each other to capture different manifestations of platelet activation. Indeed, while the percentage of platelet biomarkers relative to total particles or EVs reflects the platelet activation in ascites, PDW/PLT mirror the intravascular platelet activation. On the contrary, Pearson’s correlation analysis did not underline a correlation between platelet biomarkers percentage in ascites and PLR values in sera (r = −0.4177, *p* = 0.1274) (Figure 5B). This result suggests no linear correlation between platelet activation in ascites and systemic inflammatory state measured as platelet to lymphocytes ratio (PLR) (Figure 5C).

## 4. Discussion

There is a large body of evidence that platelets play a profound role in ovarian cancer growth and progression [37]. Previous findings demonstrated that platelets are present and active in the ascites of patients with ovarian cancer [19,38]. A recent study analysed the platelet content of fresh blood and ascites. It showed that platelets were present for the most part in an activated state within the ascites fluid in patients diagnosed with advanced-stage HGSOC [19]. However, besides this evidence, there is still the need for a method of platelet activation assessment in ascites enabling individual patient scores. Since activated platelets produce platelet-derived EVs [39], analysing these EV populations may provide additional information about the platelet status. Further, EV concentration also seems to be a potential marker of platelet activation [40].

In this study, nano-flow cytometry analyses provided quantitative information about the platelet activation status by measuring two platelet-associated markers, GPIIb/IIIa and PF4, in ascites-derived small-EVs. Since there is significant heterogeneity in the EV populations, a single marker could be able to capture only a subset of EVs. Furthermore, different markers could not be co-expressed and represent different EVs sub-population. Results indicated that although the differences were not statistically significant, the highest positivity values were obtained when both these markers were used. According to the outcomes obtained on the cohort of patients, 4.6 to 21.2% of the small-EVs stained positive for platelet biomarkers. The results could also be expressed as small-EVs positive for platelet biomarkers relative to the total number of particles, thus including the non-EV particles in the percentage. The two sets of data presented a robust correlation. These percentages were linked significantly with PDW/PLT, taken as a surrogate indicator of intravascular platelet activation [34]. This correlation is meaningful since a paracrine-based mechanism connects the platelet haemostasis with the tumour cells. Hepatic thrombopoietin synthesis that arises in response to tumour-derived IL-6 induces an increase in platelet counts, promoting tumour growth and producing a positive feedback loop [41]. Moreover, platelets were reported to extravasate into ascites in ovarian cancer patients [41]. On the contrary, we did not observe any link between the platelet activation in ascites and the systemic inflammatory state measured as platelet to lymphocytes ratio (PLR), which was likely because the inflammatory activity follows multiple pathways. According to a previous study [34], there were significant differences in FIGO stage, lymph node metastasis, and ascites between low PDW/PLT (<0.0845) and high PDW/PLT (>0.0845) groups. Our cohort’s PDW/PLT values ranged from 0.013 to 0.046, thus belonging to the low PDW/PLT range. Therefore, it will be important to test the correlation between platelet activation status in ascites and PDW/PLT in patients with high PDW/PLT values.

Malignant ascites constitute a unique tumour microenvironment that contributes to disease progression and provides a supporting structure for accumulating cells and bioactive components. It was demonstrated that the cell populations and the molecular signatures of ascites dynamically reflected the disease status [27,42]. In a previous study, we described how small-EVs isolated from ascites represented an exciting biological sample for analysing cancer-related biomarkers [27]. In light of the observation that a group of proteins (A2M, FN1, ALB, PLG, ITIH4, TF, SERPING1, FGA, SERPINA1, FGB, FGG, APOH, APOA1) were involved in platelet activation, signalling and aggregation, we considered that small EVs isolated from ascites could consequently serve as a potential platform for the study of platelet activation, which could provide specific information on the tumour microenvironment, to compare the patients and follow their progress during therapy. This measure could be a basis for investigating potential connections between the tumour microenvironment, chemoresistance and disease progression.

EVs can perform a wide range of functions in the circulation, including effects on coagulation and inflammation pathways. Moreover, these nanostructures can reach organs and tissues inaccessible to platelets, contributing to extended intercellular communication. Like EVs from other cell types, platelet-derived EVs carry various molecules such as proteins, lipids, and RNAs on their surface or within their lumen [24,25]. About 30% of all EVs in normal plasma originated from platelets, as estimated by cryo-electron microscopy [43]. As opposed to platelets, platelet EVs can cross tissue barriers, extending their activity beyond the blood [39]. In particular, platelet EVs were identified in the synovial fluid of rheumatoid arthritis patients [44,45]. Also, EVs released by platelets during states of ongoing inflammation can leave the circulation and penetrate into the bone marrow space [46]. Platelet EVs contribute to the pre-metastatic niche as part of the cooperative effort of various biochemical mediators that eventually create an enabling microenvironment in which metastatic cells can anchor and survive [47].

The separation of EVs from proteins and non-EV lipid particles in complex fluids such as ascites represents a big challenge. EV preparations are constituted by different vesicles and a greater or a lesser number of soluble proteins that may contribute to the biological activity of the fluid. Therefore, as the International Society for Extracellular Vesicles (ISEV) suggested, we must also consider the heterogeneity of the final preparations used in the different studies, including soluble factors. In this context, the most relevant term would ultimately be “EV-enriched secretome” rather than “EVs” [48]. Accordingly, AFM single-particle nanomechanical measurements performed on samples obtained via a protocol for small-EVs enrichment from ascites using precipitation with polyethylene glycol showed heterogeneous distributions in both size and mechanical stiffness of the isolated particles. While ascertaining the exact nature of individual particles is at the moment impossible, it is still possible to infer that each sample contained a mixture of co-isolated intact vesicles and non-vesicular particles from their overall nanomechanical behaviour as shown in CA/D plots (see, e.g., Figure 1B). While vesicles were found to have lower concentrations than non-EVs, it is interesting to note that they represented a large proportion of the overall particle volume due to the small sizes of non-EV particles. Another puzzling but intriguing feature revealed by CA/D plots was that the average mechanical stiffness of intact EVs enriched from different ascites samples tended to vary in different patients but remained constant in samples collected at different time points from the same patient. This suggests that ascites EVs from the same patient might have a characteristic composition which is then reflected in their mechanical properties; however, the current level of knowledge hinders the development of this conjecture from its highly speculative form at the moment.

## 5. Conclusions

Single-particle phenotyping analysis by nano-flow cytometry after labelling and ultra-centrifugation allowed us to quantify the total number of non-EV particles, the number of small-EVs and the number of platelet-derived small-EVs, providing a platelet activation assessment independent of the ascites volume and taking into account the complexity of the samples. Overall, we presented a high-throughput method that can be helpful in future studies to determine the correlation between the extent of platelet activation in ascites and disease status.

## Figures and Tables

**Figure 1 cancers-14-04100-f001:**
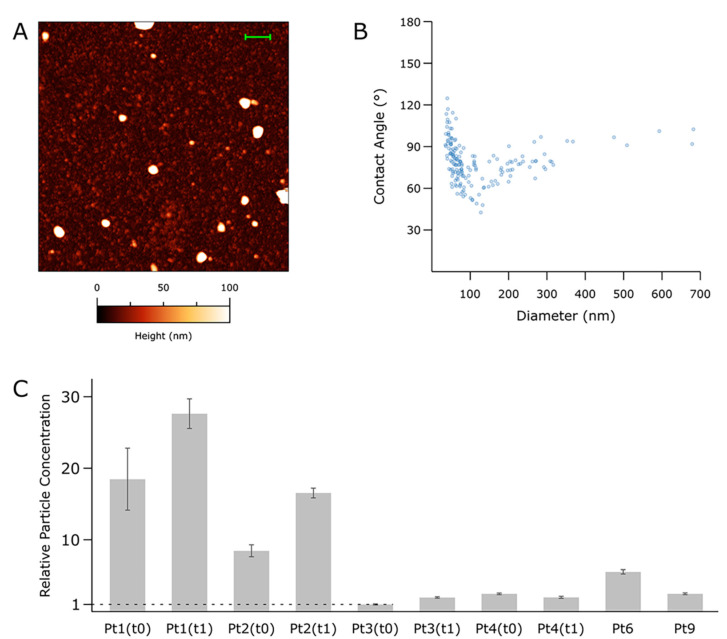
Atomic Force Microscopy (AFM)-based, single-particle nanomechanical analysis of putative vesicles found in 10 representative samples from the panel. (**A**) Representative AFM micrograph from one of the samples (Pt 6, scalebar = 1 µm). At least five such images were recorded for each sample; representative images from all the samples can be found in Appendix A. (**B**) Contact Angle (CA) vs. Diameter (D) plot for the sample shown in panel (**A**) (Pt 6). Each point represents one particle as measured in the AFM micrographs; the scatterplots resulting from each sample can be found in Appendix A. (**C**) Relative total particle concentrations as estimated via surface densities measured on multiple samples following the same deposition protocol. The vertical scale is given as multiples of the lowest concentration (Pt3, t0). Error bars are the standard deviation of densities from at least five different areas measured on the sample.

**Figure 2 cancers-14-04100-f002:**
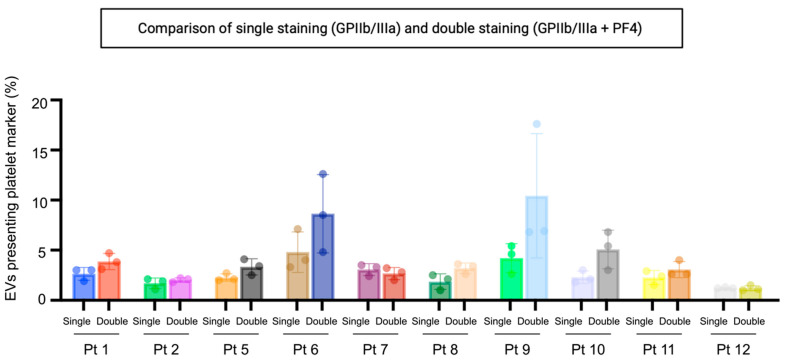
Percentage of particles positive for GPIIb/IIIa and PF4 relative to total EVs. Double labelling will identify EVs with one or both platelet markers. Samples were fluorescently labelled with FITC-conjugated antibodies specific to GPIIb/IIIa and PF4. EVs were determined as >45 nm diameter particles positive for CellMask™ deep red plasma membrane labelling. Single: single staining for GPIIa/IIIb. Double: double staining for GPIIb/IIIa and PF4. Pt: patient.

**Figure 3 cancers-14-04100-f003:**
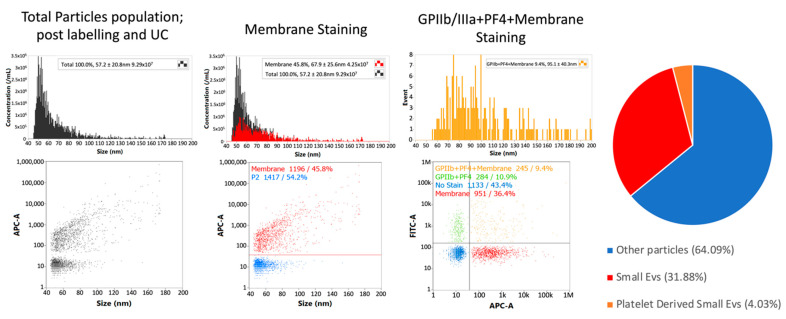
Single-particle phenotyping of ascites-derived small EVs by nFCM analysis (NanoFCM). Samples were fluorescently labelled with FITC-conjugated antibodies specific to GPIIb/IIIa and PF4 (FITC-A channel). In addition, the dye CellMask™ deep red plasma membrane stain (CM) was used (APC-A channel). Bivariate dot-plots of indicated fluorescence versus SSC are shown. The pie chart shows the average composition of all the samples analysed from the patient cohort. All percentages shown on images are for a single data point. Values from pie chart are true corrected values for all data. nFCM: nano-flow cytometry.

**Figure 4 cancers-14-04100-f004:**
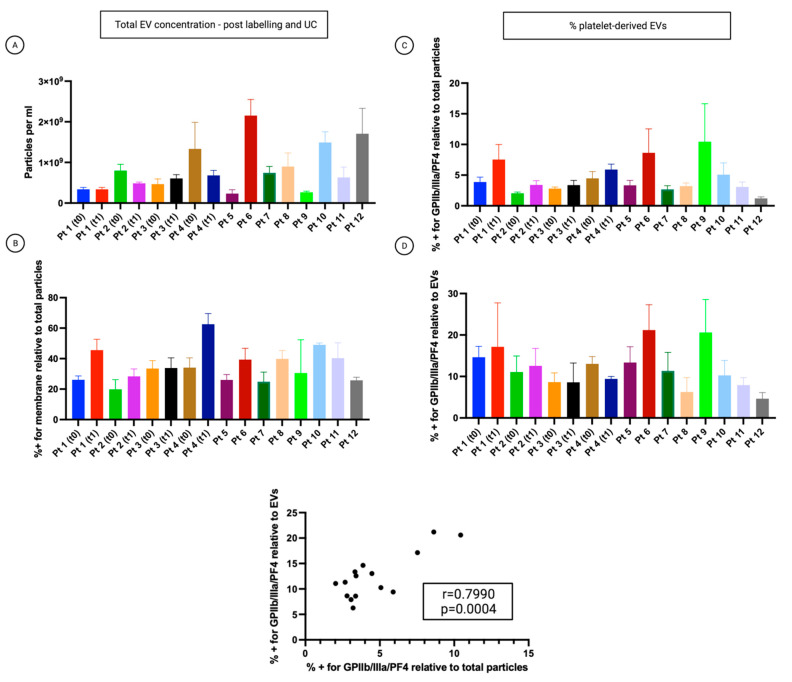
Analysis of the small-EVs from ascites on the cohort of patients. The patient’s number corresponds to Table 1. T0: time of diagnosis before starting neoadjuvant chemotherapy (NACT). T1: time of interval debulking surgery (IDS). Data are reported as means ± SD of one experiment performed in triplicate. (**A**) The concentration of all particles > 45 nm. (**B**) Percentage of all particles > 45 nm, which also showed CellMask™ labelling (thus determined as EVs). (**C**) Percentage of all particles > 45 nm, which also showed labelling for at least one platelet marker. (**D**) The percentage of CellMask™ positive particles (EVs) is also positive for one or more platelet markers.

**Figure 5 cancers-14-04100-f005:**
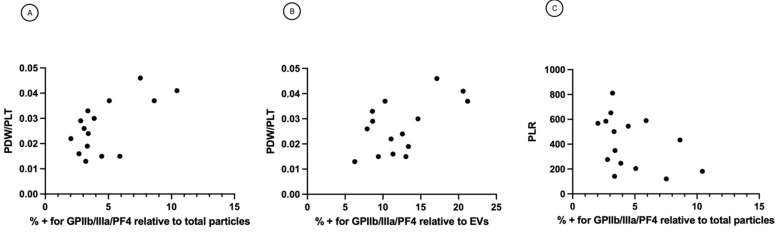
(**A**) Correlation between percentage of platelet biomarkers relative to total particles and PDW/PLT. r = 0.6307, *p* = 0.0117 (**B**) Correlation between percentage of platelet biomarkers relative to EVs and PDW/PLT. r = 0.5832, *p* = 0.0225. (**C**) Correlation between percentage of platelet biomarkers relative to total particles and PLR. r = −0.4177, *p* = 0.1274. PDW = platelet distribution width; PLT = platelet count; PLR = platelet to lymphocytes ratio.

**Table 1 cancers-14-04100-t001:** Patient selection and clinical data.

Pt	CA125 UI/mL (t0)	CA125 UI/mL (t1)	Ascites mL (t0)	Ascites mL (t1)	PDW/PLT (t0)	PDW/PLT (t1)	PLR (t0)	PLR (t1)
1	1022.4 *	443.2 *	1800	300	0.03	0.046	246.41	120
2	1851.0 *	156.9 *	5300	<100	0.022	0.024	568.18	348.6
3	2426.0 *	157.0 *	8000	<100	0.029	0.033	276.06	141.5
4	318.0 *	166.5 *	2200	5000	0.015	0.015	544.8	589.9
5	332.6 *	11.7	2000		0.019		500.98	
6	3099.1 *	12.3	4000		0.037		433.75	
7	9797.0 *	43.6	7000		0.016		584.68	
8	290.3 *	5.5	3500		0.013		811.54	
9	995.6 *	27.5	<100		0.041		181.18	
10	20,597.0 *	157.6	200		0.037		203.35	
11	2875.2 *	132.8	3700		0.026		652.54	
12	5268.3 *	38.6	4900		0.016		662.88	

Note. Pt = Patient; Reference range CA125 ≤ 25 UI/mL; t0 = Diagnosis; t1 = Post-neoadjuvant chemotherapy (post-NACT); PDW = platelet distribution width; PLT = platelet count (×103/µL—normal range 150–450); PDW = %; PLR = platelet to lymphocytes ratio. * Samples collected during laparoscopic surgery.

## Data Availability

All data supporting this study’s findings are available from the corresponding author upon reasonable request.

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
