# Peer review of "Platelet Activation in Ovarian Cancer Ascites: Assessment of GPIIb/IIIa and PF4 in Small Extracellular Vesicles by Nano-Flow Cytometry Analysis"

_cancers, 2022, doi:10.3390/cancers14174100_

Round 1
Reviewer 1 Report
The aim of this manuscript is to analyze the expression of platelet markers GPIIb/IIIa and PF4, in small – Evs populations, isolated from the ascitic fluid of patients, affected by advanced ovarian cancer.
This manuscript shows rich content, providing a deep insight for some works: I found it to be well-written. At the same time, the manuscript is original, in comparison to published literature. Even if the manuscript provides an organic overview, with a densely organized structure and based on well-synthetized evidence, there are aspects to be mentioned, to make the article fully readable. For these reasons, the manuscript requires minor changes.
Please find below an enumerated list of comments on my review of the manuscript:
INTRODUCTION:
LINE 48: This introductive section may benefit from providing an organic introduction on HGSOC, in the context of ovarian cancer. Previous studies, in fact, assessed that as the deadliest gynecological malignancy, ovarian cancer consists of a heterogeneous group of neoplasia, of which about the 90% are epithelial (mucinous, serous, endometrioid, and clear cell subtypes). In this context, high-grade serous ovarian cancer subtype (HGSOC) is more aggressive, than the low-grade serous ovarian cancers and for this reason it is characterized by a poorer prognosis. To this aim, the manuscript will benefit from providing also recent evidence, related to the origin of this aggressive form of ovarian cancer (see, for reference: Giusti, I., Bianchi, S., Nottola, S. A., Macchiarelli, G., Dolo, V. (2019). CLINICAL ELECTRON MICROSCOPY IN THE STUDY OF HUMAN OVARIAN TISSUES. EuroMediterranean Biomedical Journal, 14).
LINE 51: Before describing the pivotal role played by platelets in the tumor microenvironment of ovarian cancer, the authors should provide an organic description of the physiological role exerted by platelets. In this perspective, platelets are a cellular subgroup of the element, circulating in the bloodstream, with multiple functional roles in angiogenesis, innate immunity and cancer progression (see, for reference: Maouia A, Rebetz J, Kapur R, Semple JW. The Immune Nature of Platelets Revisited. Transfus Med Rev. 2020 Oct;34(4):209-220. doi: 10.1016/j.tmrv.2020.09.005. Epub 2020 Sep 19. PMID: 33051111; PMCID: PMC7501063). This suggestion is finalized to make the manuscript accessible, providing sufficient information for the non-expert while also achieving a balance of detail for those with more expertise in the field.
LINE 77: Although only few evidence, derived from research on clinical samples, are available to investigate…The author should reformulate this sentence in a more fluent way.
LINE 78: an-ti-platelet -> anti-platelet
Figure at page 2 (LINE 45 – 46): Please, optimize the font of this Figure, which is not so legible and easy to follow.
DISCUSSION:
LINE 339: ana – lyzed -> analyzed
LINE 340: pre-sent -> present
LINE 350: Further – more -> Furthermore
LINE 363: extrava – sate -> extravasate
LINE 374: molecu – lar -> molecular
LINE 376 and 401: bi – ological -> biological
As regards the main topic, it is interesting and certainly of great scientific and clinical impact: in fact, this manuscript touches a significant area, by analyzing the expression of platelet markers GPIIb/IIIa and PF4, in small – Evs populations, isolated from the ascitic fluid of patients, affected by advanced ovarian cancer. As regards the originality and strengths of this manuscript, this is a significant contribute to the ongoing research on this topic. Overall, the contents are rich, and the authors also give their deep insight for some works.
As regards the section of methods, there is a specific and detailed explanation for the majority of methods used in this study: this is particularly significant, since the manuscript relies on a multitude of methodological and statistical analysis, to derive its conclusions. The methodology applied is overall correct, the results are reliable and adequately discussed.
The conclusion of this manuscript is perfectly in line with the main purpose of the paper: the authors have designed and conducted the study properly. As regards the conclusions, they are well written and present an adequate balance between the description of previous findings and the results presented by the authors.
Finally, this manuscript also presents a basic structure, properly divided and characterized by organic and detailed figures and tables. This manuscript looks like very informative since there is few evidence on this topic. As regards figures, in the mentioned section, they should be optimized, as they are not legible and easy to follow.
In conclusion, this manuscript is densely presented and well organized, based on well-synthetized evidence. The authors were lucid in their style of writing, making it easy to read and understand the message, portrayed in the manuscript. Besides, the methodology design was rigorous and appropriately implemented within the study. However, many of the topics are very concisely covered. This manuscript provided a comprehensive analysis of current knowledge in this field. Moreover, this research has futuristic importance and could be potential for future research. However, minor concerns of this manuscript are with the introductive and discussive section: for these reasons, I have minor comments only for these sections, for improvement before acceptance for publication. The article is accurate and provides relevant information on the topic and I suggest minor changes to be made in order to maximize its scientific impact. I would accept this manuscript, if the comments are addressed properly.
Author Response
Point to point response
LINE 48: This introductive section may benefit from providing an organic introduction on HGSOC, in the context of ovarian cancer. Previous studies, in fact, assessed that as the deadliest gynecological malignancy, ovarian cancer consists of a heterogeneous group of neoplasia, of which about the 90% are epithelial (mucinous, serous, endometrioid, and clear cell subtypes). In this context, high-grade serous ovarian cancer subtype (HGSOC) is more aggressive, than the low-grade serous ovarian cancers and for this reason it is characterized by a poorer prognosis. To this aim, the manuscript will benefit from providing also recent evidence, related to the origin of this aggressive form of ovarian cancer (see, for reference: Giusti, I., Bianchi, S., Nottola, S. A., Macchiarelli, G., Dolo, V. (2019). CLINICAL ELECTRON MICROSCOPY IN THE STUDY OF HUMAN OVARIAN TISSUES. EuroMediterranean Biomedical Journal, 14).
We thank the reviewer for the valuable suggestions. We have now added these Paragraphs:
Ovarian cancer consists of a heterogeneous group of neoplasms, of which about 90% are epithelial (mucinous, serous, endometrioid, and clear cell subtypes) [1]; most women are diagnosed at an advanced stage of disease with a high-grade serous ovarian cancer subtype (HGSOC). HGSOC is significantly more aggressive than low-grade serous ovarian cancer and is associated with a poorer prognosis [2]. The lack of screening tools and early diagnosis methods substantially contributes to the unfavorable prognosis, suggesting a need to identify effective biomarkers for HGSOC.
The ascites is a fluid developing in the peritoneum of HGSOC patients in an advanced stage, and its cellular and extracellular vesicle (EV) composition can reveal the inflammatory environment state of the malignant disease [3]. The evidence that tumor cells release more EVs than healthy cells points to their role in tumor progression and the potential of EVs-based liquid biopsy as diagnostic biomarkers in ovarian cancer [4].
And the new References:
- Karnezis, A.N.; Cho, K.R.; Gilks, C.B.; Pearce, C.L.; Huntsman, D.G. The Disparate Origins of Ovarian Cancers: Pathogenesis and Prevention Strategies. Nat. Rev. Cancer 2017, 17, 65–74, doi:10.1038/nrc.2016.113.
- Klotz, D.M.; Wimberger, P. Cells of Origin of Ovarian Cancer: Ovarian Surface Epithelium or Fallopian Tube? Arch. Gynecol. Obstet. 2017, 296, 1055–1062, doi:10.1007/s00404-017-4529-z.
- CLINICAL ELECTRON MICROSCOPY IN THE STUDY OF HUMAN OVARIAN TISSUES. EuroMediterranean Biomed. J. 2019, 145–151, doi:10.3269/1970-5492.2019.14.34.
LINE 51: Before describing the pivotal role played by platelets in the tumor microenvironment of ovarian cancer, the authors should provide an organic description of the physiological role exerted by platelets. In this perspective, platelets are a cellular subgroup of the element, circulating in the bloodstream, with multiple functional roles in angiogenesis, innate immunity and cancer progression (see, for reference: Maouia A, Rebetz J, Kapur R, Semple JW. The Immune Nature of Platelets Revisited. Transfus Med Rev. 2020 Oct;34(4):209-220. doi: 10.1016/j.tmrv.2020.09.005. Epub 2020 Sep 19. PMID: 33051111; PMCID: PMC7501063). This suggestion is finalized to make the manuscript accessible, providing sufficient information for the non-expert while also achieving a balance of detail for those with more expertise in the field.
We thank the reviewer for the valuable suggestions. We have now added these Paragraphs:
Platelets are anucleate cell fragments formed and released into the bloodstream from bone marrow megakaryocytes (MK) [5]. Platelets serve as major contributors to hemostasis, which can be subdivided into primary hemostasis, secondary hemostasis and fibrinolysis [6]. Platelets are activated upon contact with extracellular matrix proteins, including collagen, von Willebrand factor, and fibronectin, in response to vascular injury. During primary hemostasis, platelet adhesion at a site of vascular damage entails a collaborative effort of various platelet receptors, leading to platelet activation and aggregation. In secondary hemostasis, the formation, deposition and cross-linking of insoluble fibrin, generated by the coagulation cascade, stabilize the primary platelet plug [6]. The highly regulated enzymatic process of fibrinolysis prevents unnecessary accumulation of intravascular fibrin.
Besides their hemostatic function, platelets play two major non-hemostatic roles related to the immune response: they have an anti-infective nature and influence innate and adaptive immune systems [7]. Platelets can communicate with numerous immune cells such as B-cells, T-cells, macrophages, neutrophils, EC, natural killer cells and DC, leading to a spectrum of immune-related events [7].
Platelets are recognized as pivotal players in numerous other processes, and recent literature indicates their central involvement in cancer progression and metastasis [8,9].
And the new References:
- Brass, L.F.; Diamond, S.L.; Stalker, T.J. Platelets and Hemostasis: A New Perspective on an Old Subject. Blood Adv. 2016, 1, 5–9, doi:10.1182/bloodadvances.2016000059.
- Twomey, L.; Wallace, R.G.; Cummins, P.M.; Degryse, B.; Sheridan, S.; Harrison, M.; Moyna, N.; Meade-Murphy, G.; Navasiolava, N.; Custaud, M.-A.; et al. Platelets: From Formation to Function; IntechOpen, 2018; ISBN 978-1-78985-078-9.
- Maouia, A.; Rebetz, J.; Kapur, R.; Semple, J.W. The Immune Nature of Platelets Revisited. Transfus. Med. Rev. 2020, 34, 209–220, doi:10.1016/j.tmrv.2020.09.005.
LINE 77: Although only few evidence, derived from research on clinical samples, are available to investigate…The author should reformulate this sentence in a more fluent way.
We have now reformulated the sentence:
Only a little research has been conducted on clinical samples to investigate the contribution of platelets to the biological characteristics of micrometastases. However, antiplatelet therapy was tested in patients based on the crucial contribution of platelet to the formation and expansion of the early metastatic niche [10].
LINE 78: an-ti-platelet -> anti-platelet
We apologize for the error. We have now corrected it.
Figure at page 2 (LINE 45 – 46): Please, optimize the font of this Figure, which is not so legible and easy to follow.
We apologize for the low readability of the figure. We have now enlarged the font and provided the new figure.
DISCUSSION:
LINE 339: ana – lyzed -> analyzed
We apologize for the error. We have now corrected it.
LINE 340: pre-sent -> present
We apologize for the error. We have now corrected it.
LINE 350: Further – more -> Furthermore
We apologize for the error. We have now corrected it.
LINE 363: extrava – sate -> extravasate
We apologize for the error. We have now corrected it.
LINE 374: molecu – lar -> molecular
We apologize for the error. We have now corrected it.
LINE 376 and 401: bi – ological -> biological
We apologize for the error. We have now corrected it.
As regards the main topic, it is interesting and certainly of great scientific and clinical impact: in fact, this manuscript touches a significant area, by analyzing the expression of platelet markers GPIIb/IIIa and PF4, in small – Evs populations, isolated from the ascitic fluid of patients, affected by advanced ovarian cancer. As regards the originality and strengths of this manuscript, this is a significant contribute to the ongoing research on this topic. Overall, the contents are rich, and the authors also give their deep insight for some works.
We thank the Reviewer for the comment.
As regards the section of methods, there is a specific and detailed explanation for the majority of methods used in this study: this is particularly significant, since the manuscript relies on a multitude of methodological and statistical analysis, to derive its conclusions. The methodology applied is overall correct, the results are reliable and adequately discussed.
We thank the Reviewer for the comment.
The conclusion of this manuscript is perfectly in line with the main purpose of the paper: the authors have designed and conducted the study properly. As regards the conclusions, they are well written and present an adequate balance between the description of previous findings and the results presented by the authors.
We thank the Reviewer for the comment.
Finally, this manuscript also presents a basic structure, properly divided and characterized by organic and detailed figures and tables. This manuscript looks like very informative since there is few evidence on this topic. As regards figures, in the mentioned section, they should be optimized, as they are not legible and easy to follow.
We thank the reviewer for pointing this out. We have improved the quality of the Figures, which anyhow have suffered from some drawbacks during the format conversion at the initial submission. If the problem persists, we can send the pdf files at the best resolution. We have replaced the figures, and in figure 4, we have put the insert below the graph for better readability.
In conclusion, this manuscript is densely presented and well organized, based on well-synthetized evidence. The authors were lucid in their style of writing, making it easy to read and understand the message, portrayed in the manuscript. Besides, the methodology design was rigorous and appropriately implemented within the study. However, many of the topics are very concisely covered. This manuscript provided a comprehensive analysis of current knowledge in this field. Moreover, this research has futuristic importance and could be potential for future research. However, minor concerns of this manuscript are with the introductive and discussive section: for these reasons, I have minor comments only for these sections, for improvement before acceptance for publication. The article is accurate and provides relevant information on the topic and I suggest minor changes to be made in order to maximize its scientific impact. I would accept this manuscript, if the comments are addressed properly.
We thank the Reviewer for the comments, and we have provided new elements in the introduction as suggested.
Reviewer 2 Report
In this study, authors assessed the presence of platelet activating factors in EVs to assess if it can be used to correlate with platelet activation in patient body fluids and ultimately predict the disease status. The study can be of interest in the EV field to explore EVs as a potential biomarker representing disease status. I have the following comments that need to be addressed.
Introduction last paragraph- need to add the significance of the current study
Please describe small EVs isolation protocol in detail, as isolated EVs hugely depends upon the protocol followed
How the percentage calculation was done in Figure 2 and 4 need to be explained clearly in the methods
Image resolution is very bad and some images are completely unreadable (like Figure 4C inset)
Authors are recommended to compare their particle concentration data obtained through nanoflow with nanoparticle tracking analysis. As NTA is one of the widely used and standard method for particle count, the comparison will help to standardize and validate their method.
What is the significance of assessing GPIIb/III and PF4 in EVs over direct assessment from ascites fluid?
Figure 3 is again hard to read. How did authors made sure that GPIIb/III and PF4 in indeed in EV membrane and not as a soluble proteins? Can they be present as soluble proteins? This is important to discuss as the reported precipitation-based EV isolation method can have significant soluble impurities. Authors are recommended to do co-staining EVs with fluorescent antibody targeting tetraspanin (CD63/cd81/cd9) universal marker of EV that could help to elucidate the presence of GPIIb/III and PF4 in EVs.
Author Response
Point to point response
In this study, authors assessed the presence of platelet activating factors in EVs to assess if it can be used to correlate with platelet activation in patient body fluids and ultimately predict the disease status. The study can be of interest in the EV field to explore EVs as a potential biomarker representing disease status. I have the following comments that need to be addressed.
Introduction last paragraph- need to add the significance of the current study
We thank the Reviewer for pointing this out. In response to the Reviewer's concern, we have now added a few lines in the last paragraph of the Introduction:
We herein hypothesized that platelets are activated upon extravasation in ascites and release small-EVs that can be quantified relative to the small-EVs total population by single-particle phenotyping analysis, thus enabling individual patient scores. To explore this hypothesis, we performed nano-flow cytometry analyses by measuring two platelet-associated markers, GPIIb/IIIa and PF4, in ascites-derived small-EVs. Pre-and post-chemotherapy ascites samples were obtained from the same ovarian cancer patients. Overall, our results provide for each sample the total number of non-EV particles, the number of small-EVs and the number of platelet-derived small-EVs, thus assessing the extent of platelet activation in ascites. We present a high-throughput method that can be helpful in future studies to determine the correlation between the extent of platelet activation in ascites and disease status.
Please describe small EVs isolation protocol in detail, as isolated EVs hugely depends upon the protocol followed
To respond to the Reviewer’s concern, we have now added a detailed description of the EVs isolation:
1 ml of sample (ascites fluid) was centrifuged at 2000 x g at room temperature to remove cells and debris. The supernatant containing the clarified fluid was transferred to a new tube without mixing up the pellet. 0.5 ml of Total Exosome Isolation Reagent was added. The sample was mixed by vortexing and incubated at room temperature for 30 minutes. After incubation, the sample was centrifuged at 10000 x g for 10 minutes, and the supernatant was discarded. The pellet was resuspended by adding 500 ul 1xPBS.
How the percentage calculation was done in Figure 2 and 4 need to be explained clearly in the methods
In response to the Reviewer's concern about the percentage calculation, we have now edited the Y-axis title of figure 2 to read "EVs presenting platelet marker (%)" to make it easier to understand. In figure 2 legend, we have explained that the double labelling will identify EVs with one or both platelet markers. We have changed the following sentence in the figure legend "In addition, the dye CellMask™ deep red plasma membrane stain (CM) was used" to "EVs were determined as >45nm diameter particles positive for CellMask™ deep red plasma membrane labelling".
We have added an explanation of percentage calculation in the figure 4 legend:
Figure 4. Analysis of the small-EVs from ascites on the cohort of patients. The patient’s number corresponds to Table 1. T0: time of diagnosis before starting neoadjuvant chemotherapy (NACT). T1: time of interval debulking surgery (IDS). Data are reported as means ± SD of one experiment performed in triplicate. A) Concentration of all particles >45nm. B) Percentage of all particles >45nm, which also showed CellMask™ labelling (thus determined as EVs). C) Percentage of all particles >45nm, which also showed labelling for at least one platelet marker. D) Percentage of CellMask™ positive particles (EVs) also positive for one or more platelet markers.
Image resolution is very bad and some images are completely unreadable (like Figure 4C inset)
We have seen the poor resolution of the figures, and we apologise for the inconvenience that arises somehow during the file conversion at the time of submission. If the problem persists, we can send the pdf files at the best resolution. We have replaced the figures, and in figure 4, we have put the insert below the graph for better readability.
Authors are recommended to compare their particle concentration data obtained through nanoflow with nanoparticle tracking analysis. As NTA is one of the widely used and standard method for particle count, the comparison will help to standardize and validate their method.
In response to the Reviewer's concern, we need to clarify that this paper's key interests are the fluorescently labelled particles. Whilst NTA is a commonly accessible method, it's not considered to be highly accurate or reproducible, and F-NTA does not yet have the necessary sensitivity to distinguish EVs labelled by antibodies with high fidelity. nFCM is a validated method meeting our analysis requirements; see, for reference: Vogel, R., et al. Measuring particle concentration of multimodal synthetic reference materials and extracellular vesicles with orthogonal techniques: Who is up to the challenge. Journal of Extracellular Vesicles. 10, (3), 12052 (2021).
What is the significance of assessing GPIIb/III and PF4 in EVs over direct assessment from ascites fluid?
In response to the Reviewer's concern, we need to clarify that direct assessment of these biomarkers from ascites fluids shows problems related to the presence of aggregates, impairing the quantitative analysis's robustness. Moreover, the analysis of the small-EVs population enabled us to determine a patient score, as the amount of platelet-derived EVs is related to the total number of EVs and provided quantitative data. Of interest, other biomarkers measurements can integrate this analysis, starting from the small-EVs population as a platform to investigate the tumour's biological status, as demonstrated in our previous work in Molecular Oncology.
Figure 3 is again hard to read.
We apologize for the low resolution of the Figure. We have replaced it.
How did authors made sure that GPIIb/III and PF4 in indeed in EV membrane and not as a soluble proteins? Can they be present as soluble proteins? This is important to discuss as the reported precipitation-based EV isolation method can have significant soluble impurities. Authors are recommended to do co-staining EVs with fluorescent antibody targeting tetraspanin (CD63/cd81/cd9) universal marker of EV that could help to elucidate the presence of GPIIb/III and PF4 in EVs.
The nFCM analysis was performed with SS triggering (as is common). All events measured have a SS signal indicating them to be particles >45nm. Soluble proteins do not provide this SS signal and are not included in the data output. In graph 3 the use of membrane labelling replaces the need for CD labelling as an identifier of particles with a membrane bilayer. The 3rd dot plot and size histogram under the title “GPIIb/IIIa + PF4 + membrane staining” shows events which provide three simultaneous signals “SS signal of a >45nm particle”, “Membrane labelling of an EV”, and “antibody labelling of the target protein”, providing a rigorous determination of these particles.
Regarding soluble impurities, in graph 3 the first plot and size histogram is actually of all particles.

Reviewer 3 Report
The article “Platelet activation in ovarian cancer ascites: assessment of GPIIb/IIIa and PF4 in small extracellular vesicles by nano-flow cytometry analysis” by Barbara Bortot et al. is in line with the special issue “Cancer Nanotherapy and Nanodiagnostic”.
Indeed, modern hybrid flow cytometers with nanoparticle imaging (for example, ImageStream®X MarkII and NanoAnalyzer U30 used in this article) allow good detection of small extracellular particles without binding on latex or magnetic particles, which certainly removes a number of questions related to the specificity of binding, choice adequate markers for binding, etc.
The article is methodically well executed, the research methods are presented in detail (however, in Section 2.3. it is necessary to clarify from what volume of ascites small extracellular vesicles were isolated), the vesicles are carefully characterized, the results are clearly and in detail presented. The clinical part of the work does not cause any remarks either.
The comments relate to the design of the figures (in Figure 1 B and C it is impossible to read the labels of the axes; Figure 3 is generally of poor quality, very small, fuzzy and unreadable signatures, Figure 4 - very small, fuzzy and unreadable signatures).
In the chapter “Results” it is written that “The analysis revealed that 20% to 63% of all detected particles were cell membrane positive (Figure 4B)” - what can this mean?
Conclusion. The quality of the figures needs to be improved.
Author Response
Point to point response
The article “Platelet activation in ovarian cancer ascites: assessment of GPIIb/IIIa and PF4 in small extracellular vesicles by nano-flow cytometry analysis” by Barbara Bortot et al. is in line with the special issue “Cancer Nanotherapy and Nanodiagnostic”.
Indeed, modern hybrid flow cytometers with nanoparticle imaging (for example, ImageStream®X MarkII and NanoAnalyzer U30 used in this article) allow good detection of small extracellular particles without binding on latex or magnetic particles, which certainly removes a number of questions related to the specificity of binding, choice adequate markers for binding, etc.
The article is methodically well executed, the research methods are presented in detail (however, in Section 2.3. it is necessary to clarify from what volume of ascites small extracellular vesicles were isolated), the vesicles are carefully characterized, the results are clearly and in detail presented. The clinical part of the work does not cause any remarks either.
We thank the Reviewer for their positive comments. We have now added a detailed description of small EVs isolation:
2.3. Small-EVs isolation
Small-EVs from ascites were isolated as previously described [27], using Total Exo-some Isolation Reagent (Invitrogen, CN 4484453), following the protocol specified by the manufacturers. 1 ml of sample (ascites fluid) was centrifuged at 2000 x g at room temperature to remove cells and debris. The supernatant containing the clarified fluid was transferred to a new tube without mixing up the pellet. 0.5 ml of Total Exosome Isolation Reagent was added. The sample was mixed by vortexing and incubated at room temperature for 30 minutes. After incubation, the sample was centrifuged at 10000 x g for 10 minutes, and the supernatant was discarded. The pellet was resuspended by adding 500 ul 1xPBS.
The comments relate to the design of the figures (in Figure 1 B and C it is impossible to read the labels of the axes; Figure 3 is generally of poor quality, very small, fuzzy and unreadable signatures, Figure 4 - very small, fuzzy and unreadable signatures).
We have seen the poor resolution of the figures, and we apologise for the inconvenience that arises somehow during the file conversion at the time of submission. If the problem persists, we can send the pdf files at the best resolution. We have replaced the figures, and in figure 4, we have put the insert below the graph for better readability.
In the chapter “Results” it is written that “The analysis revealed that 20% to 63% of all detected particles were cell membrane positive (Figure 4B)” - what can this mean?
This means that the percentage of all particles >45nm, which also showed CellMask™ labelling (thus determined as EVs), was between 20% and 63%. We have now clarified the analysis in the legend of figure 4 to make it easier to understand:
Figure 4. Analysis of the small-EVs from ascites on the cohort of patients. The patient’s number corresponds to Table 1. T0: time of diagnosis before starting neoadjuvant chemotherapy (NACT). T1: time of interval debulking surgery (IDS). Data are reported as means ± SD of one experiment performed in triplicate. A) Concentration of all particles >45nm. B) Percentage of all particles >45nm, which also showed CellMask™ labelling (thus determined as EVs). C) Percentage of all particles >45nm, which also showed labelling for at least one platelet marker. D) Percentage of CellMask™ positive particles (EVs) also positive for one or more platelet markers.
Conclusion. The quality of the figures needs to be improved.
We have improved the quality of the figures.

Round 2
Reviewer 2 Report
The authors have addressed my comments appropriately.